



# A method for random uncertainties validation and probing the natural variability with application to TROPOMI/Sentinel5P total ozone measurements

Viktoria F. Sofieva[1], Hei Shing Lee[1,2], Johanna Tamminen[1], Christophe Lerot[3], Fabian Romahn[4], Diego G. Loyola[4]

[1]Finnish Meteorological Institute, Helsinki, Finland
[2]University of Helsinki, Atmospheric Sciences Department, Finland
[3]BIRA, Brussels, Belgium
[4]German Aerospace Centre (DLR), Remote Sensing Technology Institute, Oberpfaffenhofen, Germany

*Correspondence to*: Viktoria F. Sofieva (viktoria.sofieva@fmi.fi)

**Abstract**

In this paper, we discuss the method for validation of random uncertainties in the remote sensing measurements based on evaluation of the structure function, i.e., root-mean-square differences as a function of increasing spatio-temporal separation of the measurements. The limit at the zero mismatch provides the experimental estimate of random noise in the data. At the same time, this method allows probing the natural variability of the measured parameter. As an illustration, we applied this
method to the clear-sky total ozone measurements by TROPOMI/Sentinel-5P.

We found that the random uncertainties reported by the TROPOMI inversion algorithm, which are in the range 1-2 DU, agree well with the experimental uncertainty estimated by the structure function.

Our analysis of the structure function has shown the expected results on total ozone variability: it is significantly smaller in the tropics compared to mid-latitudes. At mid-latitudes, ozone variability is much larger in winter than in summer.
The ozone structure function is anisotropic (being larger in latitudinal direction) at horizontal scales larger than 10-20 km. The structure function rapidly grows with the separation distance. At mid-latitudes in winter, the ozone values can differ by 5 % at separations 300-500 km.

The discussed method is a powerful tool in experimental estimates of the random noise in data and studies of natural variability and it can be used in various applications.



## 1 Introduction

The information about uncertainties of measurements are important in many data analyses: data averaging, comparison, data assimilation etc. The uncertainties are usually categorized into "random" and "systematic" (for more discussion, see von Clarmann et al. , 2020).

For remote sensing measurements, the random component of uncertainty budget is estimated via propagation of measurement noise through the inversion algorithm (e.g., Rodgers et al., 2000). If the linear/linearized model is adequate, the Gaussian error propagation in the linearized model can be used for simplicity. However, the uncertainty estimates given by an inversion algorithm might be incomplete: this might be due to incomplete/simplified models of the processes that describe the satellite measurements or/and unknown/unresolved atmospheric features. Other contributing factor might be the imperfect estimates of measurement uncertainties, as well as the uncertainties of external auxiliary data. Therefore, validation of theoretical uncertainty estimates is needed for remote-sensing measurements.

For atmospheric measurements specifically, validation of random uncertainty estimates is not a trivial task because the measurements are performed in a continuously changing atmosphere.

This short paper is dedicated to a simple method, which allows simultaneous probing small-scale variability on an atmospheric parameter and validation of random uncertainties in the measurements of this parameter.

The paper is organized as follows. Section 2 briefly describes the methodology of the analysis. In Section 3, we describe the TROPOMI total ozone data, which are used in our paper. In Section 4, we briefly explain the technical details of the computation of the structure function using TROPOMI data. The results and discussion are presented in Section 5. Summary (Section 6) concludes the paper.

## 2 Methodology

In our work, we will exploit the concept of the structure function $f(\mathbf{r})$, which characterizes the degree of spatial dependence of a random field (or a stochastic process [e.g., *Tatarskii*, 1961]:

$$D(\boldsymbol{\rho}) = D(\mathbf{r}_1 - \mathbf{r}_2) = \frac{1}{2} \left\langle \left[ f(\mathbf{r}_1) - f(\mathbf{r}_2) \right]^2 \right\rangle, \tag{1}$$

where $\mathbf{r}_1$ and $\mathbf{r}_2$ are two locations and $\boldsymbol{\rho} = \mathbf{r}_1 - \mathbf{r}_2$. This concept assumes that the random field is locally homogeneous, which is the spatial equivalence of a random process with stationary increments. In spatial statistics, $D(\boldsymbol{\rho})$ is called the variogram (Wackernagel, 2003).



When using experimental data, the difference of ozone in two locations is defined not only by the natural variability but also by uncertainty of measurements. Therefore, with the spatial separation $\rho \to 0$ , $D(\boldsymbol{\rho})$ tends to the random uncertainty variance $\sigma_{noise}^2$ .

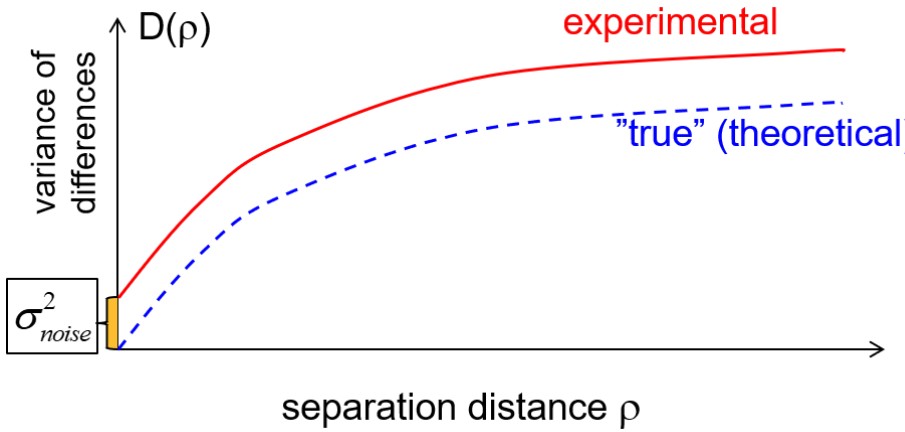

**Figure 1. The schematic representation of the structure function estimated from atmospheric measurements.**

The application of this method requires many measurement points with different spatial and temporal separations, including very small separations. For satellite measurements in limb-viewing geometry, such information is very limited. Nevertheless, several applications using this method have been published. Staten and Reichler (2009) applied this method to the validation of radio-occultation measurements by Constellation Observing System for Meteorology, Ionosphere, and Climate (COSMIC), which consists of identical instruments on board of six microsatellites. In their work, the authors evaluated two-dimensional structure function using the data from the beginning of COSMIC mission, when the satellites were in close orbits (and therefore measurements in close separations were found). An analogous method - evaluation of the 1D structure function in polar regions (with transformation of temporal mismatch to spatial separation) - has been applied for validation of random uncertainty estimates of the MIPAS (Michelson Interferometer for Passive Atmospheric Sounding) and GOMOS (Global Ozone Monitoring by Occultation of Stars) instruments on board the Envisat satellite (Laeng et al., 2013; Sofieva et al., 2014). The 1D structure function has been evaluated in (Sofieva et al., 2008) using collocated temperature profiles by radiosondes at Sodankylä.

For satellite measurements in nadir-viewing geometry, the smallest separation is usually defined by the ground pixel size of an instrument, and the application of the structure function method looks very attractive: measurements with small spatio-temporal mismatch can be found in all locations and in all seasons. However, we are not aware of the application of the structure function method for validation of random uncertainty estimates from nadir-viewing satellite instruments. In our paper, we use total ozone measurements by TROPOMI on board Sentinel-5P, which has a very fine spatial resolution, for the illustration of the structure function method.




## 3    Case study: total ozone data by TROPOMI

The TROPOspheric Monitoring Instrument (TROPOMI) satellite instrument on board the Copernicus Sentinel-5 Precursor (S5P) satellite was launched in October 2017 (http://www.tropomi.eu; https://sentinel.esa.int/web/sentinel/missions/sentinel-5p, Veefkind et al., 2012) . The mission of S5P is to perform atmospheric measurements with high spatio-temporal resolution for monitoring air quality and forecasting climate. TROPOMI implements passive remote sensing techniques by measuring the solar radiation reflected, scattered and radiated by the Earth/atmosphere system at ultraviolet, visible, near-infrared and

shortwave infrared wavelengths in the nadir-looking geometry. With a large spectral range covered, TROPOMI data allows to measure vertical columns for a wide number of atmospheric gases, including ozone ($O_3$), nitrogen dioxide ($NO_2$), sulphur dioxide ($SO_2$), carbon monoxide (CO), methane ($CH_4$), and formaldehyde (HCHO), with an extremely good spatial resolution (3.5 x 5.5 km² since August 2019). This allows applying the structure function method, since the ground pixel separations can be probed at very small scales.

The data are available in near-real-time, offline and reprocessing streams. In our studies, the Level 2 offline data product of total ozone column (TOC) is used. This product relies on the operational implementation of the GODFITv4 algorithm, used for producing total ozone climate data records from many nadir-viewing sensors (GOME, SCIAMACHY, GOME-2, OMI, OMPS) with excellent performance (Garane et al., 2019; Lerot et al., 2014). Total ozone columns are derived using a non-linear minimization procedure of the differences between measured and modelled sun-normalized radiances in the ozone

Huggins bands (fitting window: 325-335 nm).

The total ozone product includes an estimate for the random uncertainty associated to each observation. The latter is simply obtained by the propagation through the inversion solver of the radiance and irradiance statistical errors provided with the measurements (in Level 1 products). In addition to the instrumental noise, some pseudo-random errors (i.e. systematic errors varying rapidly at short spatio-temporal scales) may be present in the data due to imperfect corrections for the presence of

clouds in the probed scene. In order to limit this term, our analysis will focus on the clear-sky condition only. We use the operational TROPOMI cloud product (Loyola et al., 2018) to select ozone data with cloud fraction smaller than 0.2.

Figure 2 shows typical TROPOMI clear-sky total ozone column observations in one orbit. Typical values of random uncertainties (Figure 2, center) range from 0.5 to 2 DU. As shown in Figure 2 (right), the measurement points in a certain latitude band are performed nearly at the same time, for one orbit.






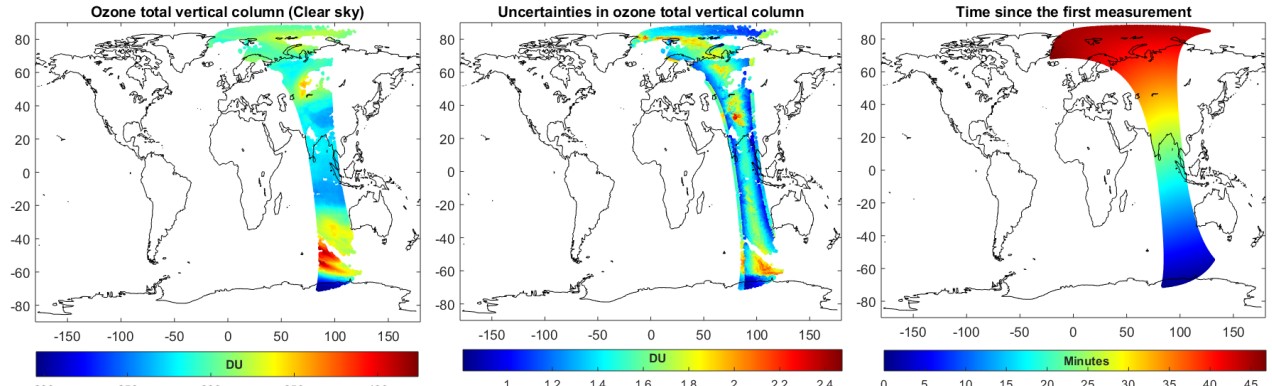

**Figure 2 Left: TROPOMI clear-sky ozone measurements in one orbit (September 1, 2018, 06:37), Center: uncertainty. Right: Time since the first measurement in this orbit**

## 4    Evaluation of ozone structure functions using TROPOMI data

In our analyses, we selected the TROPOMI Level 2 clear-sky total ozone data in several broad latitude bands (60-90S, 30-60S, 20S-20N, 30-60N, 60-90 N) and in certain months: July 2018, October 2018, January 2019, and March 2019. We computed the structure function for each latitude band and for each month. In order to exclude the temporal dependence, we evaluated the structure functions for each orbit separately, and then average over a month. In our work, we evaluate two-dimensional structure function, i.e., the variance of ozone differences as a separation in latitude and in longitude.

The computation of structure function requires finding the differences in ozone and the corresponding spatial separation (i.e. distance in latitude and longitude) between every pair of data pixels. Theoretically it could be achieved by considering one point and comparing it with the rest of observations, then moving to another point and again comparing it with all other observations. However, owing to the very high spatial resolution of TROPOMI and thus an extremely large amount of observation points even for one orbit, such operation is very demanding computationally. To ensure numerical efficiency, the

algorithm is simplified while preserving the underlying principle: instead of using all observations we consider sufficiently large amount of observations. For each orbit and for each latitude band, we create a set of ~100 reference points evenly spatially spaced in a selected latitude zone. For each reference point, we computed differences from all points in the latitude zone in both horizontal and vertical directions. This operation allows collecting many pairs of data corresponding to all separation distances (2-2.5 million).

After computing the average of squared difference in ozone and spatial separation for each orbit, the monthly-averaged structure functions are created. The monthly averaged is based on 400-450 structure functions from individual orbits, so in total 800-1000 million data pairs are used for evaluation of monthly averaged structure functions.



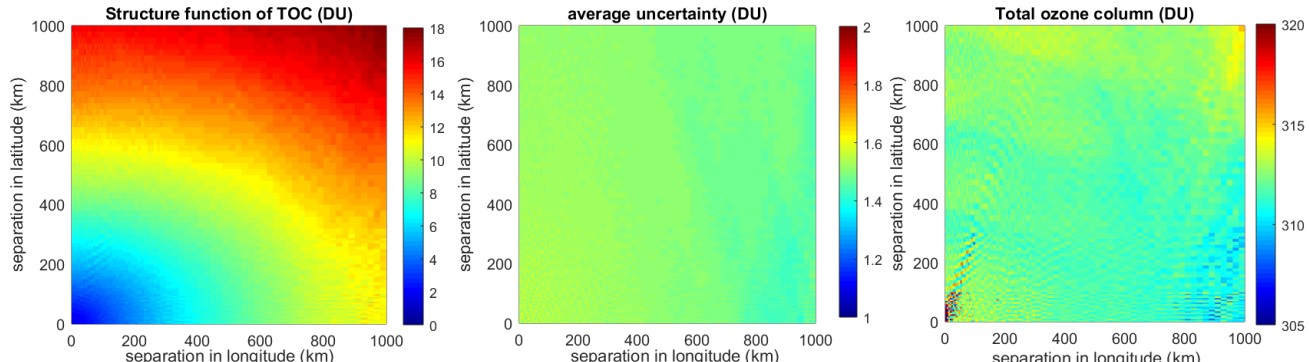

**Figure 3 Illustration of structure function in July 2018 and other associated parameters, for latitude 30- 60 N. Left: the structure function expressed as $\sqrt{D(\rho)}$ (DU), Center: mean uncertainty (DU) corresponding to the separations (pairs of points). Right: mean ozone column (DU) corresponding to the separations.**

Figure 3 (left) shows the example of the structure function evaluated for July 2018 in the latitude band 30-60N. As expected, the rms of the ozone differences grows with increasing separation distance. The structure function is anisotropic: it is larger in latitudinal direction. In the selected latitude band (this is also the case for other months and latitude bands), the mean error estimate corresponding to different separation distances is nearly constant (~1.5 DU, Figure 3, center). Analogously, the mean total ozone corresponding to different separation distances is also nearly constant (Figure 3, right). This implies that the structure function looks similar in both absolute (DU) and relative (%) representations (see also below).

## 5   Results and discussion

The structure functions evaluated in different latitude bands and seasons are shown in Figure 4. Color represents $\sqrt{D(\rho)}$ expressed in DU. An analogous figure showing the structure function in relative units (%) is presented in Figure 6 in Appendix. As mentioned above, the structure functions in absolute and in relative units look very similar.

The obtained morphology of ozone variability is quite expected: it is overall much smaller in the tropics than at middle and high latitudes, where it has a pronounced seasonal cycle. At mid-latitudes in winter and spring, the ozone variability is very strong, even for small separations. Except at high northern latitudes in winter and spring, the structure functions are anisotropic with a stronger variability in the latitudinal direction (see also the discussion below).



**Figure 4 Structure function (in DU) for different latitude bands (columns) and months (rows).**

150     Figure 5 shows the structure function with the focus on small separations. The error estimates given by the inversion algorithm

are also presented in the figure. We observe, that in the regions of small (20S-20N) or moderate variability (30-60S and 30-60

N in local summer), the structure function approaches at zero limit nearly exactly to the theoretical random error estimates in

the data. This indicates that the random uncertainty estimates provided by the inversion algorithm are close to reality.

Interestingly, in the regions of large ozone variation (mid-latitudes in local winter), the limit of structure function at zero is





155    larger than the predicted error estimates by ~0.3-0.5 DU. This might be due to imperfectness of Level 1 error estimates, or due

to remaining natural variability at very small scales in the regions of high ozone gradients.

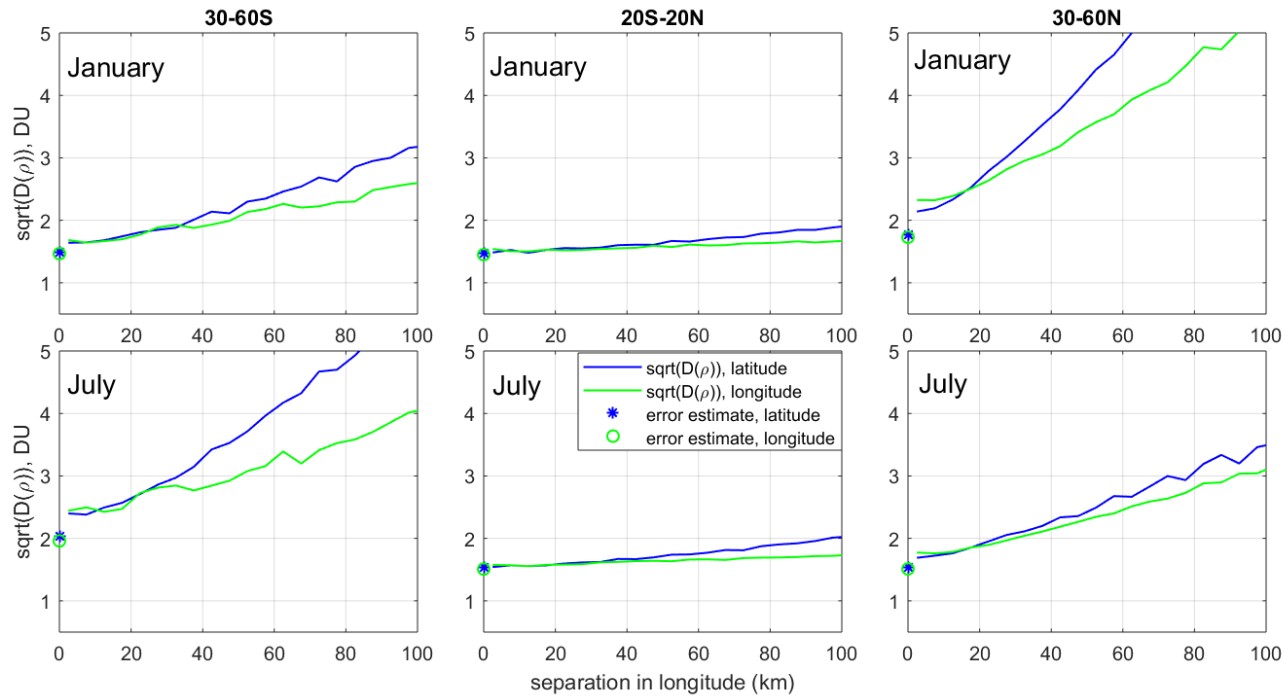

**Figure 5. Structure function (in DU) for different latitude bands (columns) and months (rows), focus on small separations.**

160

In relation to the structure of the ozone variability, one can notice also an interesting feature in Figure 5: The ozone structure

functions are nearly isotropic at small scales, below 20- 40 km, and then anisotropy grows with the increasing separation

distance. This variable anisotropy is an interesting feature, but its analysis is beyond the scope of our paper. At NH polar

regions (Figure 4), the structure function remains isotropic until large scales of 300-400 km.

165        It is quite evident that the structure function method can be applied to any dataset, in which the data with different

separation distances can be found. The approach might especially useful for other remote sensing measurements in nadir

looking geometry, which have fine horizontal resolution. The datasets should not be necessarily remote sensing measurements.

The structure function can be applied, for example, also to modelled data by a chemistry-transport model, in order to estimate

numerical noise in the model.



## 6   Summary

The analyses performed in our paper have shown that the structure function method – i.e. the evaluation of rms differences as a function of increasing spatial separation - is a powerful tool, which allows quantification of random noise in the data. The limit at zero mismatch provides the experimental estimate of the random uncertainty variance. In our paper, we applied the structure function method to validate the TROPOMI clear-sky total ozone random uncertainty estimates by the inversion algorithm. We found that the latter are very close to the experimental ones provided by the structure function method, in the regions of small total ozone natural variability. This indicates adequacy of the TROPOMI random error estimation.

At the same time, the structure function method provides the detailed information about the natural variability of the measured parameter. For TROPOMI total ozone, we have analyzed the structure functions in different seasons and latitude zones. We found the expected results: the overall variability is the smallest in the equatorial region, and the largest variability is at mid- and high latitudes in winter and spring. At these locations/seasons, the rms of ozone differences grows rapidly with the separation between measurements achieving ~5 % at distances of 300-500 km. Our analysis has shown that the structure function is anisotropic (variability is larger in the latitudinal direction) at separations of a few hundred kilometers nearly everywhere, except at northern polar regions. For lower separation distances (up to 20-40 km), the structure function generally remains isotropic.

The structure function method discussed in the paper can be equally applied to other remote sensing measurements or atmospheric model data.



## 7    Appendix



**Figure 6. Structure function (in %) for different latitude bands (columns) and months (rows).**


### Data availability

Sentinel-5 Precursor TROPOMI data are available from the Copernicus Open Access Hub at https://scihub.copernicus.eu.




*Author contributions*

VS and HSL have performed the analyses and wrote the major part of the text. CL, FR and DL are the developers of the TROPOMI total ozone inversion algorithm. All the authors contributed to manuscript writing.

*Competing interests*

The authors declare that they have no conflict of interest.

*Acknowledgements*

The authors thank EU/ESA/DLR for providing the TROPOMI/S5P Level 2 products used in this paper. The work of VS, HSL
and JT was supported by the ESA-funded project SUNLIT and the Academy of Finland, Centre of Excellence of Inverse Modelling and Imaging. The work of FR and DL for the development of TROPOMI retrieval algorithms and processors has been supported by DLR (S5P KTR 2472046). VS thanks the TUNER team for the useful discussions.

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
