# Peer review of "A method for random uncertainties validation and probing the natural variability with application to TROPOMI/Sentinel5P total ozone measurements"

_Atmospheric Measurement Techniques, 2020_

## Referee Comment (RC1) · Anonymous Referee #1 · 18 Dec 2020

The manuscript presents a method based on estimation of the nugget of the structure function (or variogram) for validation of "random" uncertainty estimates in remote sensing retrievals. The method is illustrated on TROPOMI total column ozone measurements and is used to show that the reported random uncertainty estimates are typically reasonable in regions with low-moderate variability but can underestimate the uncertainty in regions of high ozone variability. The methods proposed are a potentially very valuable tool for validating random uncertainty estimates but I have the following comments. After addressing these, I think the manuscript is worthy of acceptance.

[Figure]

Major Comments:

1. I would like to see further explanation of the methodology and its necessary assumptions, most explicitly the importance of stationarity. The paper appears to be focused on presenting a new methodology to the community that is applicable to many other areas beyond the TROPOMI analysis (which I certainly agree with), but the authors do not give a reader who is not already familiar with variogram analysis the tools to know how to apply it to another application. Overall, there is very little explanation (or references) of the structure function/variogram method, how it's estimated or it's assumptions, beyond a couple of lines in Section 2. As an example, in Section 4, the TROPOMI variogram analysis is separated into latitude bins, by month, and across orbits presumably in an attempt to satisfy stationarity assumptions, but there is no explanation to the reader of why this needs to be done in order for the variogram estimates to be meaningful.

2. Related to 1., the literature review is sparse and inclusion of additional references in spatial statistics would be extremely useful for any reader who intends to use the methodology. Examples of such references include, for general variogram analysis:

Matheron, G. (1963). "Principles of geostatistics". Economic Geology. 58 (8): 1246–1266.

Cressie, N., 1993, Statistics for spatial data, Wiley Interscience

And for methods involving estimation of the nugget effect see for example:

Kang, E. L., Cressie, N., and Shi, T. (2010), "Using temporal variability to improve spatial mapping with application to satellite data,"Canadian Journal of Statistics, 38, 271–289.

3. The TROPOMI analysis focuses only on clear-sky conditions due to the fact that "some pseudo-random errors (i.e. systematic errors varying rapidly at short spatio-temporal scales) may be present in the data due to imperfect corrections for the presence of clouds in the probed scene." For validation of the propagation of only measurement error uncertainty I can see why this is necessary, but wouldn't it also be a powerful use of the method to show/find if the presence of other errors (e.g. from non-clear sky conditions) result in an underestimate of uncertainty? I.e. if the nugget estimate is substantially higher than that reported by the algorithm?

Minor Comments:

1. Define all acronyms the first time they are used, e.g. TROPOMI, rms, etc.

2. Pg2. Line 32: It would be helpful to define explicitly what is meant by random vs systematic error here.

3. Pg. 3, line 61: "The application of this method requires many measurement points with different spatial and temporal separations, including very small separations." – the methodology as presented ignores temporal correlation, so only small spatial separations are needed.

4. Section 4: How did you decide upon these spatial and temporal bins, is stationarity reasonable here?

5. Pg. 5, line 124: Replace "horizontal and vertical directions" with longitude and latitude

6. Section 4: By computing structure function estimates from each orbit separately, you have an ensemble from which you compute a final mean estimate. Why not also look at the variability information from the ensemble when assessing if the nugget is consistent with that reported by the TROPOMI algorithm? Standard deviations or quantiles of the structure function estimated from the ensemble would provide further information about how consistent the nugget estimate is with that reported from the algorithm.

7. Figure 3, center: shrink the color scale to the value range (1.4-1.6ish)

8. Figure 3, right: I am not sure exactly what is being computed here or what information this plot provides. Is the mean being taken over all of the points that are included in the differences taken in each bin? In that case, almost all of the data should be included in each lat/long bin except at very large lags. This would mean that the averages in each bin are taken over mostly the same data and should be consistent?

9. Figure 5: Does the TROPOMI inversion algorithm provide a footprint level uncertainty estimate? If so, is the single value used in figures an average of these estimates within lat bin/month? Please provide further explanation.

---

## Referee Comment (RC2) · Anonymous Referee #2 · 4 Jan 2021

**Review of the paper « A method for random uncertainties validation and probing the natural variability with application to TROPOMI/Sentinel5P total ozone measurements » by V. Sofieva et al**

The paper presents a method of validation of random component of uncertainty estimates in remote sensing measurements, using the 2-dimensional structure functions, and illustrates it on TROPOMI ozone total column data. The paper is clearly written, is straight to the point, and I strongly recommend it for the publication in AMT, provided the following minor issues will be addressed.

Detailed comments:

- Lines 53-55: the sentence "this concept assumes that the random field is locally homogeneous, which is the spatial equivalence of a random process with stationary increments" is misleading: the notion of the structure function exists for any random process, not necessarily for those with stationary increments. In return, for the real-valued process with stationary increments, the structure function is one of its two main characteristics (see for example Yaglom, 1987 or Kolmogorov, 1940). I would first instroduce $D(\rho)$ via Eq. 1, call it variogram and would give the reference to Wackernagel for details, than would explain its link to the structure functions, which are already covered by previous papers of the first author.
- Fig. 1. The red curve touching the $\rho=0$ line is misleading: if in eq.1 one writes $\rho1=\rho2$, the $D(\rho)$ is zero. Either the line should stop shortly before touching the coordinate line, or you should precise that the value of the estimated variability at $\rho=0$ is obtained by prolongation by continuity. The first solution would keep in line with general concise style of the paper, the second would be compliant with the formulation of your summary.
- Line 98: it would be better to align the term "pseudo-random error" with the terminology of (von Clarmann et al, 2020)
- Lines 123-124: what is the minimal separation distance of your sample, and of which size is corresponding subsample?

Language / formulation comments :

- line 31: are - > is
- line 36 : " … in the linearized model" can be thrown away
- Line 67: "1D" better to be written in words.
- line 101: "… to select ozone data…" -> "in which we select ozone data …"
- line 134: detail the abbreviation "rms"
- line 166: "… might especially BE useful…"

References:

Kolmogorov, A. N.: Wiener's spiral and some other interesting curves in Hilbert space, Dokl. Akad. Nauk SSSR, 26, 115–118, 1940.

Yaglom, A. M.: Correlation Theory of Stationary and Related Random Functions, Volume I: Basic Results, Springer, New York, 1987.

---

## Author Comment (AC1) · 2 Feb 2021

Dear Reviewer,

Thank you very much for your positive evaluation and comments on our manuscript. We took your comments into account in the revised version of the manuscript. Please find below our detailed replies (black font) on your comments (blue font).

Major Comments:
1. I would like to see further explanation of the methodology and its necessary assumptions, most explicitly the importance of stationarity. The paper appears to be focused on presenting a new methodology to the community that is applicable to many other areas beyond the TROPOMI analysis (which I certainly agree with), but the authors do not give a reader who is not already familiar with variogram analysis the tools to know how to apply it to another application. Overall, there is very little explanation (or references) of the structure function/variogram method, how it's estimated or it's assumptions, beyond a couple of lines in Section 2. As an example, in Section 4, the TROPOMI variogram analysis is separated into latitude bins, by month, and across orbits presumably in an attempt to satisfy stationarity assumptions, but there is no explanation to the reader of why this needs to be done in order for the variogram estimates to be meaningful.

2. Related to 1., the literature review is sparse and inclusion of additional references in spatial statistics would be extremely useful for any reader who intends to use the methodology. Examples of such references include, for general variogram analysis:
Matheron, G. (1963). "Principles of geostatistics". Economic Geology. 58 (8): 1246–1266.
Cressie, N., 1993, Statistics for spatial data, Wiley Interscience

and for methods involving estimation of the nugget effect see for example:

Kang, E. L., Cressie, N., and Shi, T. (2010), "Using temporal variability to improve spatial mapping with application to satellite data," Canadian Journal of Statistics, 38, 271–289.

Authors:
Thank you for the suggestion and references. In the revised version, we explain the methodology in more detail, and added these and other relevant references in Section 2. In Section 4, we explained the selection of latitude zones and the analysis in more detail.

Reviewer #1
3. The TROPOMI analysis focuses only on clear-sky conditions due to the fact that "some pseudo-random errors (i.e. systematic errors varying rapidly at short spatiotemporal scales) may be present in the data due to imperfect corrections for the presence of clouds in the probed scene." For validation of the propagation of only measurement error uncertainty I can see why this is necessary, but wouldn't it also be a powerful use of the method to show/find if the presence of other errors (e.g. from non-clear sky conditions) result in an underestimate of uncertainty? I.e. if the nugget estimate is substantially higher than that reported by the algorithm?

Authors:
Yes, the same method can be used for detecting and characterization of "pseudo-random" errors. In the revised version, we added a new figure comparing the structure functions and the uncertainty estimates cloud-free and cloudy conditions.

Minor Comments:
1. Define all acronyms the first time they are used, e.g. TROPOMI, rms, etc.

Authors: Corrected

2. Pg2. Line 32: It would be helpful to define explicitly what is meant by random vs systematic error here.

Authors: In the following paragraph, we explained the origin of random errors (which are the subject of our paper) in remote sensing satellite data. We believe that reference to the comprehensive overview of ( von Clarmann et al., 2020) is sufficient here.

3. Pg. 3, line 61: "The application of this method requires many measurement points with different spatial and temporal separations, including very small separations." – the methodology as presented ignores temporal correlation, so only small spatial separations are needed.

Authors:
In principle, small spatial and temporal separations are needed. In our analysis, nearly instantaneous measurements are used for evaluation of the structure functions (this is the consequence of measurement principle by sun-synchronous satellites). This is discussed in more detail in the revised version.

4. Section 4: How did you decide upon these spatial and temporal bins, is stationarity reasonable here?

Authors:
We would like to note that only stationary increments are assumed in evaluation of the structure function. In the revised version, we explained the selection of latitude zones (natural ozone variability is expected to depend on latitude and season).

5. Pg. 5, line 124: Replace "horizontal and vertical directions" with longitude and latitude

Authors: Corrected.

6. Section 4: By computing structure function estimates from each orbit separately, you have an ensemble from which you compute a final mean estimate. Why not also look at the variability information from the ensemble when assessing if the nugget is consistent with that reported by the TROPOMI algorithm? Standard deviations or quantiles of the structure function estimated from the ensemble would provide further information about how consistent the nugget estimate is with that reported from the algorithm.

In principle, this idea is interesting. However, the estimates of the structure from one orbit are significantly less accurate than those for one month. However, the ensemble of structure function values at very small separations can be considered, as well as the ensemble of the corresponding uncertainty estimates. In the revised version, we replaced Figure 5 with the analogous, but two-dimensional structure functions at small separations (in order to illustrate how the ensemble is collected), and added a new figure with the statistical plots of experimental uncertainty estimates ( from structure function values at small separations, ex-poste) and the uncertainty estimates by the inversion algorithm (ex-ante).

7. Figure 3, center: shrink the color scale to the value range (1.4-1.6ish)

Authors:
Such color scale is chosen purposely, in order to demonstrate that the the mean error estimate corresponding to different separation distances is nearly constant.

8. Figure 3, right: I am not sure exactly what is being computed here or what information this plot provides. Is the mean being taken over all of the points that are included in the differences taken in each bin? In that case, almost all of the data should be included in each lat/long bin except at very large lags. This would mean that the averages in each bin are taken over mostly the same data and should be consistent?

Figure 3 (right) shows the mean ozone value in the pairs corresponding to different separation distances. As expected, it is nearly constant. In the revised version, we indicate this.

9. Figure 5: Does the TROPOMI inversion algorithm provide a footprint level uncertainty estimate? If so, is the single value used in figures an average of these estimates within lat bin/month? Please provide further explanation.

In the revised version, Figure 5 is replaced with the two-dimensional analogous figure, in which mean uncertainties corresponding small separation distances are also shown. They are monthly average in the corresponding latitude zone, for the corresponding to the separation distance. This is clarified in the revised version.

---

## Author Comment (AC2) · 2 Feb 2021

Dear Reviewer,

Thank you very much for your very positive evaluation and comments on our manuscript. We took your comments into account in the revised version of the manuscript. Please find below our detailed replies on your comments.

Reviewer#2

Lines 53-55: the sentence "this concept assumes that the random field is locally homogeneous, which is the spatial equivalence of a random process with stationary increments" is misleading: the notion of the structure function exists for any random process, not necessarily for those with stationary increments. In return, for the real-valued process with stationary increments, the structure function is one of its two main characteristics (see for example Yaglom, 1987 or Kolmogorov, 1940). I would first instroduce $D(\rho)$ via Eq. 1, call it variogram and would give the reference to Wackernagel for details, than would explain its link to the structure functions, which are already covered by previous papers of the first author.

Authors:

Thank you for your suggestion. We modified the paper according to your suggestion and added more references and explanations (according to comments by Reviewer #1).

Reviewer #2
- Fig. 1. The red curve touching the $\rho=0$ line is misleading: if in eq.1 one writes $\rho_1=\rho_2$, the $D(\rho)$ is zero. Either the line should stop shortly before touching the coordinate line, or you should precise that the value of the estimated variability at $\rho=0$ is obtained by prolongation by continuity. The first solution would keep in line with general concise style of the paper, the second would be compliant with the formulation of your summary.

Authors: We have corrected the figure by separation of the red curve from zero. In the paper text, we mentioned that there is a discontinuity at zero.

Reviewer #2
- Line 98: it would be better to align the term "pseudo-random error" with the terminology of (von Clarmann et al, 2020)

Authors: In the revised version, we noted that such errors belong to "model errors" in the terminology of (von Clarmann et al., 2020).

Reviewer#2
- Lines 123-124: what is the minimal separation distance of your sample, and of which size is corresponding subsample?

Authors: The smallest bin for evaluation of the structure function is 5x5 km$^2$, and the corresponding sub-sample contains over 14 000 pairs. This information is added in the revised version of the paper.

Reviewer #2
Language / formulation comments:

- line 31: are - > is

- line 36 : " … in the linearized model" can be thrown away

- Line 67: "1D" better to be written in words.

- line 101: "… to select ozone data…" -> "in which we select ozone data …"

- line 134: detail the abbreviation "rms"

- line 166: "… might especially BE useful…"

Authors:  Corrected.